# Cardiac Metastases in Neuroendocrine Neoplasms: A Single-Center Experience of Clinical Characteristics and Outcomes

**DOI:** 10.3390/cancers17243907

**Published:** 2025-12-06

**Authors:** Raphaela D. Lewetag, Nils F. Trautwein, Monika Zdanyte, Jonas Mück, Patrick Krumm, Ulrich M. Lauer, Stephan Singer, Bence Sipos, Christian la Fougère, Lars Zender, Clemens Hinterleitner, Martina Hinterleitner

**Affiliations:** 1Department of Medical Oncology and Pneumology (Internal Medicine VIII), University Hospital Tuebingen, 72076 Tuebingen, Germany; 2ENETS Center of Excellence, University Hospital Tuebingen, 72076 Tuebingen, Germanychristian.lafougere@med.uni-tuebingen.de (C.l.F.); 3Department of Nuclear Medicine and Clinical Molecular Imaging, University Hospital Tuebingen, 72076 Tuebingen, Germany; 4Department of Cardiology, Angiology and Cardiovascular Medicine, University Hospital Tuebingen, 72074 Tuebingen, Germany; 5Department of Diagnostic and Interventional Radiology, University Hospital Tuebingen, 72076 Tuebingen, Germany; 6DFG Cluster of Excellence 2180 ‘Image-Guided and Functional Instructed Tumor Therapy’, University of Tuebingen, 72076 Tuebingen, Germany; 7German Cancer Consortium, German Cancer Research Center, 72076 Tuebingen, Germany; 8Department of Pathology, University Hospital Tuebingen, 72076 Tuebingen, Germany; 9Cancer Biology and Genetics, Memorial Sloan Kettering Cancer Center, New York, NY 10065, USA

**Keywords:** neuroendocrine neoplasm, cardiac metastasis, positron emission tomography (PET/CT), somatostatin receptor imaging, peptide receptor radionuclide therapy (PRRT)

## Abstract

Cardiac metastases (CM) from neuroendocrine neoplasms (NEN) are rare and often clinically silent, leading to underdiagnosis. While metastatic spread in NEN is a known prognostic factor, the clinical significance of cardiac involvement remains poorly understood. Multimodal imaging, particularly somatostatin receptor positron emission tomography/computed tomography (SSTR PET/CT), improved detection, but the prevalence, clinical characteristics, treatment impact, and outcomes associated with CM are not well defined. . By investigating cardiac metastases as a time-dependent covariate, our analysis offers a nuanced assessment of their influence on therapeutic decision-making. We demonstrate that arrhythmia burden does not correlate with the number of cardiac lesions, indicating that lesion count alone is an insufficient marker of electrophysiologic risk. Moreover, the use of consistent SSTR PET/CT imaging across the cohort enables an accurate diagnosis of cardiac metastases, helping to address the under-recognition reported in earlier studies that relied on heterogeneous imaging modalities.

## 1. Introduction

Neuroendocrine neoplasms (NEN) represent a rare and heterogeneous entity of solid tumors, comprising 0.5–2.0% of all newly diagnosed malignancies [1]. These neoplasms are classified into two distinct groups based on their biological characteristics: well-differentiated neuroendocrine tumors (NET) and poorly differentiated neuroendocrine carcinomas (NEC) [2]. NEN can originate from various anatomical sites, with the gastrointestinal tract including the pancreas (>60%) and the lung (>20%) being the most common locations [3]. At the time of primary diagnosis, metastases are observed in approximately 50% of all cases [4], predominantly affecting lymph nodes and liver [5]. In metastatic neuroendocrine neoplasms, both tumor burden and metastatic distribution significantly influence patient outcomes [6,7]. Notably, a high liver tumor volume independently predicts a shorter progression-free survival (PFS) in grade 1 gastroenteropancreatic neuroendocrine tumors (GEP-NETs G1), while the presence of concomitant extrahepatic metastases markedly worsens overall survival OS [6]. Complementary analysis of SEER data from 2010 to 2014 corroborated that the site of metastasis itself serves as an independent prognostic factor, with liver and brain metastases associated with the poorest survival outcomes [7]. However, cardiac metastatic burden was not analyzed in these studies [6,7].

Cardiac metastases (CM) in solid tumors tend to be underdiagnosed, due to their typically subclinical or asymptomatic presentation [8]. In cancer-related autopsy series, cardiac involvement was reported in up to 2.3–18.3% [9]. The most common tumor entities causing CM include pleural mesothelioma, lung cancer, breast cancer, hematological malignancies, sarcomas, and melanomas, with the latter showing cardiac spread in more than half of the metastatic cases in postmortal autopsies [8,9,10,11]. CM primarily involve the pericardium and myocardium, with dissemination occurring via hematogenous and lymphatic routes, intracavitary seeding, or direct local invasion [9].

In contrast, CM in NEN are exceedingly rare, with an estimated prevalence of 1.5% among all NEN [12]. These metastases are most frequently located in the left ventricle, followed by the right ventricle, pericardium, and ventricular septum [12]. Carcinoid heart disease, driven by serotonin-mediated fibrosis that causes progressive thickening, immobility, and dysfunction of cardiac valves, differs fundamentally from direct metastatic invasion of the myocardium. The clinical presentation of CM in NEN varies widely, ranging from asymptomatic cases to those with overt heart failure [12,13]. Multimodal diagnostic approaches are needed to diagnose cardiac metastasis in NEN patients including transthoracic echocardiography (TTE), computed tomography (CT), cardiac magnetic resonance imaging (cardiac MRI), and somatostatin receptor positron emission tomography/computed tomography (SSTR PET/CT) [12,13].

Given the low number of patients with cardiac metastases in NEN, no international guidelines exist. Therefore, therapeutic decisions are frequently made individualized within a multidisciplinary setting. In addition, clinical relevance of CM in NEN, as well as its influence on PFS and OS, has not been conclusively clarified. This retrospective single-center study aims to characterize clinical characteristics and outcomes of NEN patients with CM.

## 2. Materials and Methods

### 2.1. Patient Characteristics

Between January 2017 and May 2025, a total of 1201 NEN patients were diagnosed and/or treated at ENETS Center of Excellence Tuebingen, Germany. We retrospectively analyzed all patients for presence of CM. Therefore, multimodal imaging results, such as echocardiography, CT, cardiac MRI, and SSTR PET/CT were screened using predefined keywords (“cardiac”, “ventricle”, “atrium”, and “septum”). In cases with ambiguous findings, image reinterpretation was performed in consultation with nuclear medicine, radiology, or cardiology. In total, 15 patients with CM in NEN could be identified. Figure 1 demonstrates the patient selection procedure. The study was approved by the IRB (ethics committee of the Faculty of Medicine of the Eberhard Karls University Tuebingen) of the University Hospital Tuebingen and was conducted in accordance with the Declaration of Helsinki (reference number 227/2025BO2). Data collection in these 15 patients contained patients data and tumor characteristics including sex, age in years at first diagnosis of NEN, TNM classification at first diagnosis of NEN, grading and Ki67 index at first diagnosis, primary tumor sites, and symptoms of CM as well as previous medical history and treatment details from the electronic patient chart, and laboratory values including chromogranin A (CgA), neuron-specific enolase (NSE), and lactate dehydrogenase (LDH). Patient characteristics are shown in detail in Table 1.

### 2.2. Imaging and CM Detection

Imaging of the identified 15 patients included TTE, CT, cardiac MRI, and SSTR PET. One patient had CM first suspected on CT and confirmed by cMRI; another symptomatic patient was diagnosed by cMRI after echocardiography. The remaining 13 patients were diagnosed de novo by SSTR PET/CT. Four patients underwent cMRI, and in three of them the lesion number and location matched SSTR PET/CT findings. In two cases, cMRI preceded PET/CT, and in one, PET/CT preceded cMRI. The single patient initially evaluated with echocardiography received an external cMRI-based diagnosis of a solitary cardiac lesion, which was no longer detectable on SSTR PET/CT after two PRRT cycles; no follow-up cMRI was performed. Eight patients had thoracic or whole-body contrast CT, but only three showed cardiac lesions. Overall, SSTR PET/CT detected CM in 14 of 15 patients, with the only missed case being the patient described above. If obtainable, data were evaluated for number and anatomical localization of cardiac metastases, cardiac molecular tumor volume, and total molecular tumor volume. The molecular tumor volume (MTV) was performed by a threshold-based semiautomatic volumetric segmentation using the software tool Affinity Hybrid Viewer (version 7.2.0, Hermes Medical Solution, Sweden), as previously described [14]. Pathologic SSTR expression was defined as standardized uptake value (SUV), which was higher than 1.5 times the mean SUV of the liver plus 2 times the standard deviation (SDliver):*MTV* = *SUVtumor* > 1.5 × *SUVmeanliver* + 2 × *SDliver*

First, a semiautomatic “single click segmentation” was performed to identify all volumes of interest with an SUV higher than the reference SUV. These areas were afterwards selected and reviewed by a trained nuclear medicine physician, who excluded physiological SSTR-expressing areas (e.g., kidney and pituitary gland) as well as non-disease-related lesions.

### 2.3. PET Acquisition and Imaging Analysis

Two different tracers were used for SSTR-PET imaging [18F]SiTATE and [68Ga]HA-DOTATATE, as described previously [15,16]. Scans with [68Ga]Ga-HA-DOTATATE were performed on a short axial field of view scanner (Biograph mCT; Siemens (Forchheim, Germany) Healthineers; acquisition continuous bed motion of 0.7 mm/s) 20 min after i.v. injection of 3 MBq/kg BW of the tracer. Scans with [18F]SiTATE were conducted either on a short axial field of view scanner (Biograph mCT; Siemens Healthineers; acquisition continuous bed motion of 0.7 mm/s) or on a long axial field of view scanner (Siemens Biograph Vision Quadra; Siemens Healthineers; acquisition time of 5 min per bed position) 90 min after i.v. injection of 2–3 MBq/kg BW of the tracer. For the Siemens Biograph Vision Quadra PET reconstruction was performed according to the standard clinical reconstruction protocol, with an Ordinary-Poisson Ordered-Subsets Expectation-Maximization algorithm (OP-OSEM 4 iterations, 5 subsets; Gaussian filter 4 mm), as described previously [17]. The Biograph mCT data were corrected for attenuation as well as scatter and reconstructed iteratively with OSEM3D (2 iterations, 21 subsets; Gaussian filter, 2 mm). The definition of pathologic SSTR expression was performed in joint consensus of a board-certified nuclear medicine physician and a physician with several years of experience in hybrid imaging.

### 2.4. PRRT Treatment Modalities

PRRT was administered in accordance with the practical guidelines jointly issued by the European Association of Nuclear Medicine and the Society of Nuclear Medicine and Molecular Imaging, following the NETTER-1 protocol [18,19]. Each treatment cycle consisted of an intravenous injection of 7400 MBq ± 10% [177Lu]Lu-HA-DOTATATE or [177Lu]Lu-DOTATATE, accompanied by an amino acid infusion for renal protection. One patient received two cycles of PRRT with 3000 ± 10% MBq [90Y]Y-DOTATOC, likewise accompanied by an amino acid infusion for renal protection. Overall, patients underwent between 2 and 8 PRRT cycles. [177Lu]Lu-HA-DOTATATE was prepared according to good manufacturing practice and the German Medicinal Products Act (AMG § 13 2b). [177Lu]Lu-DOTATATE was provided by Novartis [7]. In 2 patients, a radiosensitizer regimen was administered, as previously described [16]. Across all available structured reports, PRRT was tolerated as well, and no treatment discontinuations or hospitalizations attributable to toxicity were recorded. Reported hematologic toxicities comprised four cases of grade 1 anemia, two cases of grade 1 leukopenia, two cases of grade 1 neutropenia, one episode of grade 1 thrombocytopenia, and two episodes of grade 2 thrombocytopenia. Non-hematologic toxicity was limited to a single case of grade 1 creatinine elevation, without subsequent renal deterioration.

### 2.5. Statistical Analysis

Descriptive statistics were applied to characterize patients according to sex, age, TNM classification, histopathology including Ki67 index at first diagnosis of NEN, primary tumor site, treatment schemes in the first line, and in further therapy lines. For continuous variables, Student’s t- or Mann–Whitney U tests were used. OS and PFS, including the median, were calculated using the Kaplan–Meier method. All statistical tests were considered statistically significant if *p* was below 0.05. Statistical analysis was performed using GraphPad Prism (v.10.6.0).

## 3. Results

In total, 1201 NEN patients, diagnosed and/or treated at ENETS Center of Excellence Tuebingen, Germany, between January 2017 and May 2025, were analyzed. Fifteen of these patients were identified with CM, comprising a prevalence of 1.25% of CM in NEN (Figure 1). CM diagnosis was predominantly metachronous (93%), with only one synchronous case.

The cohort comprised eight male (53.3%) and seven female (46.7%) patients (Table 1). The age at first diagnosis of NEN was 56.1 ± 16.4 years (range: 23–83 years, n = 15) (Table 1). Primary tumor sites were predominantly the ileum (n = 12, 80.0%), followed by the lung (n = 2, 13.3%), and one single case of NEN located in the stomach (n = 1, 6.7%) (Figure 2A and Table 1). Tumor grading in these patients was distributed between G1 (n = 8, 53.3%), followed by G2 (n = 5, 33.3%), and G3 (n = 2, 13.3%) (Figure 2B and Table 1). Ki67 index at first diagnosis ranged from less than 1% to 25% with a mean of 7.7% ± 8.6 SD (Figure 2C and Table 1). TNM classification based on the 8th edition of the American Joint Committee on Cancer (AJCC) staging system at primary diagnosis is shown in Figure 2D and Table 1. The majority of these patients presented with lymph node metastases at timepoint of primary diagnosis (n = 11, 73.3%) (Figure 2E,F). Of note, only five patients showed distant metastases at timepoint of primary diagnosis of NEN.

Serum biomarkers frequently used in NEN at CM diagnosis were not determined in all patients. Compared to primary diagnosis, CgA levels were slightly higher at timepoint of CM diagnosis (Median: 138 vs. 230 μg/L). However, due to high variability and small patient numbers, this trend was not statistically significant (*p* = 0.2998). NSE (*p* = 0.2693) as well as LDH (*p* = 0.9337) levels showed no significant differences between primary diagnosis and diagnosis of CM (Appendix A). To further assess the relationship between cardiac-specific biomarkers and CM, we examined BNP and troponin levels in both symptomatic and asymptomatic patients and analyzed their associations with CM localization and MTV values (Appendix A). Although this analysis has the potential to provide additional insights into the interplay between CM and cardiac function, the limited availability of biomarker data and the small patient cohort unfortunately prevented more definitive conclusions.

CM were most frequently identified via SSTR PET/CT. In only two cases, CM was diagnosed via cardiac MRI after suspected TTE or CT. Figure 3 shows exemplarily CM of 3 patients in SSTR PET/CT, cardiac MRI, and transthoracic or transesophageal echocardiography.

CM were located most frequently in the ventricles, with the left ventricle involved in 39.3% of the cases, the right ventricle in 18.2%, and the ventricular septum in 15.2%. Atrial invasion was found less frequently (27.3%) (Figure 4A). In total, 8 out of 15 patients showed several CM with a maximum of five metastases in 1 patient (mean CM per patient 2.2, range 1–5). After diagnosis of CM, 4 out of 15 patients (26.7%) showed symptoms including atrial fibrillation (n = 2), angina (n = 1), and severe tricuspid regurgitation as well as right heart failure (n = 1) in a case where a metastasis was located at the tricuspid valve annulus leading to a post-interventional right heart failure after a gastrostomy surgery (Figure 4B–D). Total MTV ranged from 0.04 to 3120 mL, whereas MTV in the cardiac metastases ranged from 0.08 to 12.5 mL. Compared to patients without symptoms of CM, symptomatic patients showed comparable total and CM-specific MTV (Figure 4E,F). Levels of B-type natriuretic peptide (BNP), troponin, and creatinine kinase (CK) as biomarkers for heart failure and myocardial damage were measured only in some patients and at different timepoints during the course of the disease. Electrocardiogram (ECG) was performed in 9 out of 15 patients at different timepoints during the course of the disease and for different reasons. In these nine patients, repolarization disorder was seen in three cases, as well as bundle branch blocks (right bundle branch block n = 1, left anterior fascicular block n = 1) and QTc prolongation (n = 1).

All therapy lines are demonstrated in Figure 5D and Table 2. Being the backbone of NEN treatment, 12 out of 15 patients (80.0%) were treated with somatostatin analogs (SSA) during first diagnosis of CM (Table 2). Two patients were on postoperative cancer surveillance program, and one was diagnosed with NEN synchronic to CM. After CM detection, therapeutic regime changed in 11 out of 15 patients (73.3%), via expanding the treatment regime with PRRT in 8 of the 11 patients (72.7%). In the remaining patients, chemotherapeutic regime was adjusted in one patient with capecitabine/temozolomide (CAPTEM) and in another with carboplatin/etoposide (CARBO/ETO). In one patient being on follow-up, SSA therapy was initiated (Figure 5E and Table 2). No local therapeutic options, such as irradiation or surgery of the CM, were performed. Overall, seven of these patients did benefit from therapeutic regime change (63.6%), with five of them receiving PRRT. Among these seven patients, one patient achieved partial remission following PRRT, while the remaining six maintained stable disease (n = 6). Two patients deceased after the regime change, one following PRRT and the other after exposure to CARBO/ETO.

In this cohort of 15 patients, the median follow-up was 110 months (range 4–271). At data cut-off, six patients (40.0%) deceased. Time to CM from first diagnosis of NEN ranged from 0 to 207 months (median 54 months) (Figure 5A). Median OS from CM diagnosis was 95 months, with an estimated 5-year survival rate of 77%, while median OS from NEN diagnosis was 271 months, with an estimated 5-year survival rate of 87% (Figure 5B,**C**).

## 4. Discussion

In this retrospective single-center analysis, we analyzed 15 patients with CM in NEN with a focus on clinical characteristics, diagnosis, therapy, and outcome. Prevalence of CM in NEN was 1.25% in our cohort with primary sites including ileum, lung, and stomach. Most patients harbored well-differentiated G1/G2 tumors with low-to-moderate Ki67 index. We observed that cardiac involvement was most often a metachronous event, frequently detected incidentally during follow-up imaging via SSTR PET/CT scans. CM in NEN are typically clinically silent with symptoms in 26.7% of our patients. In these cases, clinical symptoms varied between atrial fibrillation, angina pectoris, tricuspid valve regurgitation, as well as heart failure in one case. Anatomical distribution pattern favored a ventricular location, with left ventricular metastases being most frequent. Although MTV combined with chromogranin has been established as a prognostic marker in NENs [14], with only four symptomatic patients, no associations could be identified between symptoms and CM localization, number, or MTV, though such relationships, and corresponding OS differences, may emerge in a larger cohort. Therapeutic changes were performed in 73.3% of the patients after first diagnosis of CM, with PRRT being a common adjustment to ongoing SSA therapy. Although CM were present, OS remained favorable, with an estimated 76.6% after 5 years.

In line with previous findings, prevalence of CM in NEN was exceedingly rare [12]. As previously described, we also saw that the primary tumor sites of patients with CM were located in the small intestine favoring the ileum [12,13], followed by bronchial NEN [20,21,22]. To this moment, there are only two cases of CM from confirmed pancreatic primary sites. Additionally, 10 cases of primary NEN of the heart are described in the literature, summarized in a combined case report and literature review in 2021 [23]. However, in these cases, SSTR PET/CT was not frequently performed [23,24,25] and histopathological findings after surgery were not described in detail [23,25,26]. As a result, a definitive allocation to a confirmed primary cardiac NEN remains vacant, especially as the heart does not normally contain neuroendocrine cells.

As CM in NEN often are clinically silent [13,27], diagnosis remains challenging. Early detection via imaging, especially with SSTR PET/CT, seems to be essential for detection of CM. As a result, the majority of the CM cases in NEN are diagnosed via SSTR PET/CT [28,29,30]. With high spatial resolution, cardiac MRI might provide additional details on the anatomical localization and is commonly used as a complementary imaging technique [28,29,30]. Echocardiography has a limited role in identifying CM but remains useful in assessing valvular heart disease in patients with carcinoid syndrome [28,29,31].

CM are most frequently diagnosed metachronously as a late manifestation of advanced systemic disease. Liu et al. reported a median interval of 62 months from initial diagnosis to cardiac involvement, consistent with Arnfield et al., who likewise observed CM only during disease progression rather than at baseline [13,31]. Our data is in line with these previous findings.

SSTR PET/CT in NEN not only assesses the presence of metastatic spread, but it is also needed to distinguish which patients might benefit from PRRT [28]. PRRT plays a feasible and promising role in the treatment of NEN and also in the treatment of CM. In another single-institution retrospective study of 25 NEN patients with CM, PRRT was applied in over half (64%) of patients [13]. Patients who underwent PRRT exhibited a tendency toward extended median OS (76 vs. 14 months) compared to those who did not receive PRRT, although this difference did not achieve statistical significance, likely due to limited cohort size (13). In line with our data, patients receiving PRRT after diagnosis of CM did benefit from therapeutic regime change. There was an observed trend of longer median OS among the group of n = 8 patients receiving PRRT (271 vs. 181 months), and PFS appeared to be prolonged in those receiving PRRT following the initial diagnosis of CM (58 vs. 19 months). However, this trend also did not reach statistical significance, presumably due to the small cohort size. PRRT was overall well tolerated with no adverse effects on cardiac function or arrhythmias during treatment [13,32]. Specific treatments, including surgical resection of the cardiac mass or localized radiotherapy, preserve a high risk, and due to the frequent lack of clinical symptoms, these are rather used exceptionally [27,33]. As most of those patients present with concurrent metastasis on other anatomical sites, systemic treatment is favored [27].

In this single center experience, median OS as well as 5-year survival remained beneficial, in line with previous case series reporting that cardiac involvement in NEN is frequently indolent and does not necessarily translate into a poor prognosis [34]. While CM may raise concerns regarding arrhythmia, heart failure, or sudden cardiac death, the clinical course in most patients appears to be driven primarily by the overall tumor burden, proliferative activity, and extent of extra-cardiac disease [34,35].

The mechanisms underlying the indolence of CM in NEN remain incompletely elucidated. The heart offers a unique microenvironment due to its mechanical, metabolic, and immunological properties. The continuous contraction, shear stress, and cyclic strain subject metastatic cells to an extreme mechanical milieu [36]. In addition, the myocardium extracts approximately 60–80% of the oxygen delivered, leaving minimal reserve capacity; thus, increased demands are predominantly met only through augmented coronary blood flow [37]. This results in an environment that is highly perfused yet metabolically constrained, promoting the survival of small intramyocardial metastases while restricting their substantial growth. From an immunological side, cardiac tissue is characterized by specialized tissue-resident macrophage populations that maintain homeostasis and facilitate repair, yet exhibit relatively subdued inflammatory responses under physiological conditions [38]. These macrophage populations are currently under investigation and may influence metastatic infiltration of the heart. Overall, these environmental constraints are reflected in the clinical observation that CM typically remain small, intramyocardial, and clinically asymptomatic, with symptomatic obstructive lesions being rare. Beyond these microenvironmental influences, we further hypothesize that CM in NEN typically grow slowly and infiltratively, replacing myocardial tissue in a diffuse and non-compressive pattern without necessarily inducing ischemia. Also, the tumor cells themselves are electrically inert, as they lack the excitable ion-channel repertoire of cardiomyocytes (e.g., fast voltage-gated sodium channels and inward-rectifier potassium channels) [39]. In addition, the myocardium exhibits substantial conduction redundancy through a dense and highly organized gap junction network, which allows electrical activation to bypass non-conductive tumor-infiltrated regions, as long as transition zones remain smooth and continuous [40,41]. These features stand in clear contrast to the well-characterized arrhythmogenic substrate of myocardial fibrosis. Fibrosis, especially when caused by infarction, creates abrupt conduction barriers that are highly arrhythmogenic even when spatially limited [40,41]. Post-infarction border zones contain heterogeneous mixtures of surviving myocytes interspersed with fibrosis, creating intermediate electrophysiological properties with slow-conducting channels and regions of unidirectional block that facilitate re-entrant arrhythmias [40,41,42]. Fibrotic remodeling also increases myocardial stiffness and impairs diastolic relaxation, whereas it can be hypothesized that infiltrative CM generally do not compromise coronary perfusion or ventricular compliance as much until very advanced disease stages. This might provide plausible explanations for the predominantly silent clinical behavior of CM in NEN. The predilection of CM for the left ventricle may reflect a predominantly hematogenous rather than lymphatic mechanism of spread. Hematogenous cardiac dissemination is described in other tumor entities such as melanoma, lymphoma, and sarcoma, and is associated with myocardial and endocardial metastases [43]. In this context, the frequent infiltration of the left ventricle may be explained by its high myocardial perfusion, dense capillary network, and greater myocardial mass, which together may facilitate hematogenous microvascular tumor lodging and intramyocardial infiltration.

This study is a single-center experience regarding characteristics and outcomes of CM in NEN. Expected limitations are due to the small cohort size and the retrospective study design. Notably, the observed 1.25% prevalence of CM in this cohort should be interpreted cautiously, since not all 1201 patients received all three imaging modalities (SSTR PET/CT, cardiac MRI, and echocardiography). Standardized imaging and systematic biomarker acquisition in prospective multicentric registries could refine risk stratification and clarify whether early systemic intervention after CM detection can translate to improved outcomes.

## 5. Conclusions

CM in NEN are rare, with only 15 cases out of 1202 patients identified in this single-center study. The use of SSTR PET/CT proved to be a crucial diagnostic tool for detecting CM in NEN. PRRT emerged as a potentially effective treatment strategy for CM in NEN, with eight patients receiving PRRT following CM diagnosis and a total response to therapeutic regime change achieved in five patients (63.6%). Despite the presence of cardiac metastases, OS remained favorable in this small cohort of NEN patients. We therefore hypothesize that cardiac metastases may be an indicator of metastatic spread rather than a standalone diagnostic determinant of survival. Future studies with larger, multicenter cohorts are warranted to validate these findings and to better define the clinical implications of CM in NEN.

## Figures and Tables

**Figure 1 cancers-17-03907-f001:**
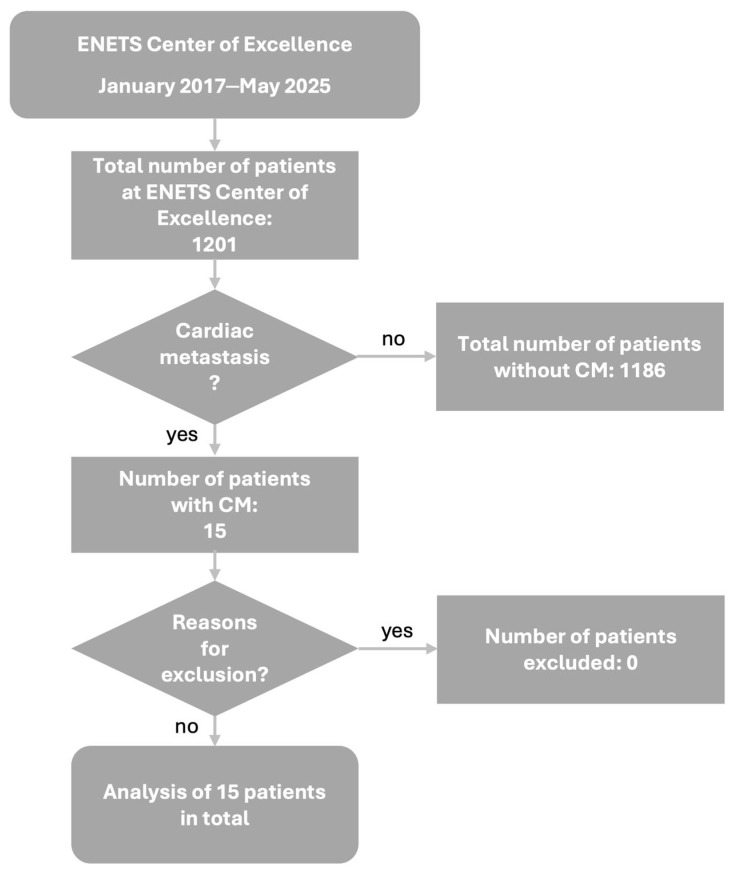
Patient selection algorithm. Abbreviations: CM = cardiac metastases, ENETS = European Neuroendocrine Tumor Society.

**Figure 2 cancers-17-03907-f002:**
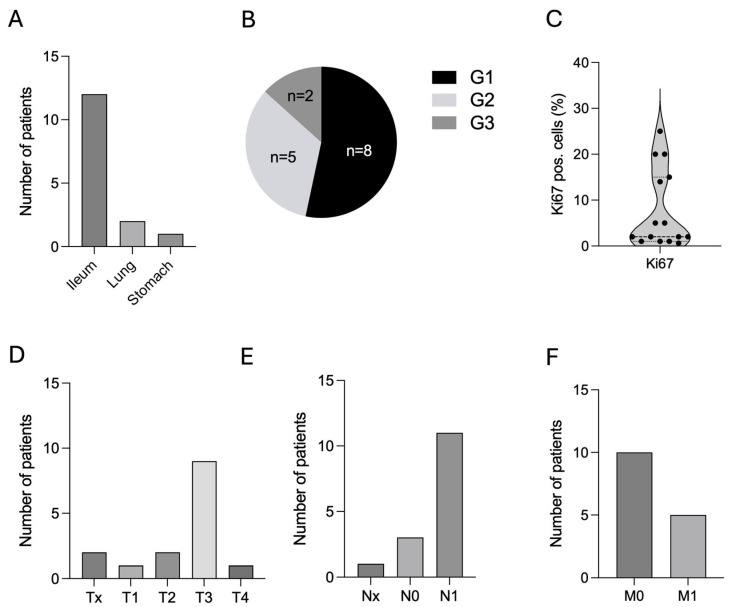
Patients’ characteristics. (**A**). Primary sites of NEN patients with CM, n = 15. (**B**). Histological grading according to WHO classification (G1–G3), n = 15. (**C**). Ki67 index in %, n = 15. (**D**–**F**). TNM classification, n = 15. Abbreviations: CM = cardiac metastases, G = grading, n = number, NEN = neuroendocrine neoplasm, TNM = tumor/nodes/metastases, WHO = World Health Organization, % = percentage.

**Figure 3 cancers-17-03907-f003:**
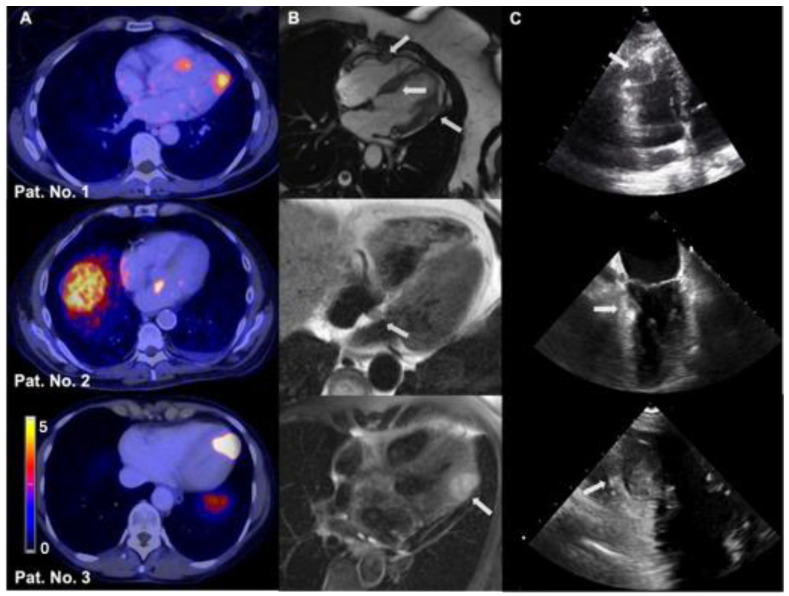
Representative multimodal imaging examples of cardiac metastases in 3 NEN patients. (**A**). SSTR PET/CT. (**B**). Cardiac MRI. (**C**). Echocardiography (patient no. 1 and no. 3 received transthoracic echocardiography, patient No. 2 underwent transesophageal echocardiography). The shown cardiac tumor lesions are marked by gray arrows, respectively. Abbreviations: CM = cardiac metastases, MRI = magnetic resonance imaging, NEN = neuroendocrine neoplasm, No = number, SSTR PET/CT = somatostatin receptor positron emission tomography/computed tomography.

**Figure 4 cancers-17-03907-f004:**
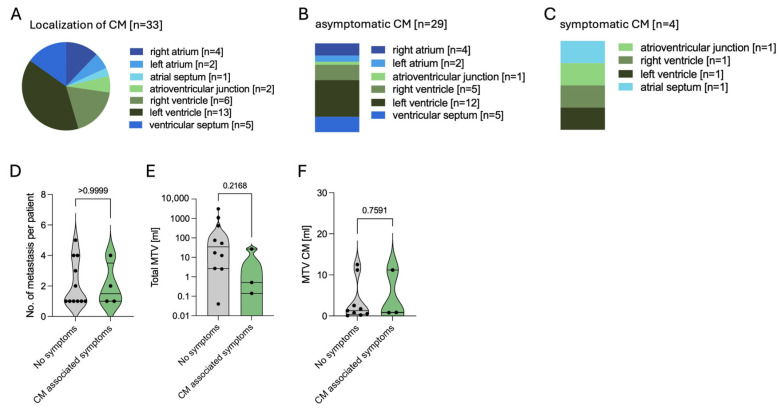
Characteristics of cardiac metastases in NEN. (**A**). Anatomical localization of all n = 33 cardiac metastases. (**B**). Localization of n = 29 asymptomatic lesions. (**C**). Localization of n = 4 symptomatic lesions. (**D**). Number of cardiac metastases per patient in asymptomatic vs. symptomatic cases. (**E**). Total molecular tumor volume (MTV) in symptomatic and asymptomatic patients concerning CM. (**F**). MTV of cardiac metastases in symptomatic and asymptomatic patients. Abbreviations: CM = cardiac metastases, MTV = molecular tumor volume, n = number, NEN = neuroendocrine neoplasm, vs. = versus.

**Figure 5 cancers-17-03907-f005:**
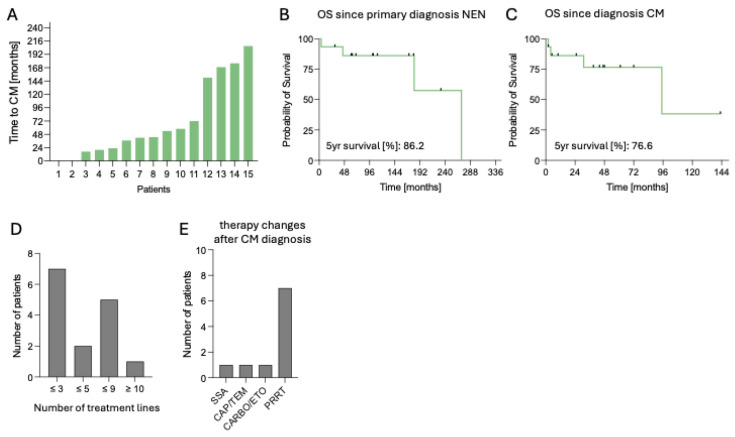
Clinical course and outcomes of patients with cardiac metastases from neuroendocrine neoplasms. (**A**). Time from primary NEN diagnosis to diagnosis of CM in months, n = 15. (**B**). OS since primary NEN diagnosis, n = 15. (**C**). OS since CM diagnosis. (**D**). Distribution of patients according to number of treatment lines (≤3, ≤5, ≤9, ≥10), n = 15. (**E**). Post-CM treatment regimens, including SSA, CAPTEM, CARBO/ETO, or PRRT, n = 15. Abbreviations: CAPTEM = capecitabine/temozolomide, CARBO/ETO = carboplatin/etoposide, CM = cardiac metastases, n = number, NEN = neuroendocrine neoplasm, OS = overall survival, PRRT = peptide receptor radionuclide therapy, SSA = somatostatin analog.

**Table 1 cancers-17-03907-t001:** Baseline patient characteristics of the study cohort (*n* = 15). Variables include sex, age at first diagnosis, TNM classification according to AJCC 8th edition, tumor grading, Ki67 index, and primary tumor site. Values are presented as absolute numbers with corresponding percentages unless otherwise specified. Abbreviations: AJCC = American Joint Committee on Cancer, n = number, NEN = neuroendocrine neoplasm, SD = standard deviation, % = percentage.

Patient Characteristics	Total (*n* = 15)
**Sex, n (%)**	
male	8 (53.3)
female	7 (46.7)
Age in years at first diagnosis of NEN	
mean ± SD (range)	56.1 ± 16.4 (23, 25, 45, 46, 52, 56, 56, 58, 59, 64, 65, 67, 68, 74, 83)
TNM classification at first diagnosis of NEN	
Tumor, n (%)	
Tx	2 (13.3)
T1	1 (6.7)
T2	2 (13.3)
T3	9 (60.0)
T4	1 (6.7)
Node, n (%)	
Nx	1 (6.7)
N0	3 (20.0)
N1/2	11 (73.3)
Metastases, n (%)	
M0	10 (66.7)
M1	5 (33.3)
Grading, n (%)	
G1 (Ki67 > 3%)	8 (53.4)
G2 (Ki67 3–20%)	5 (33.3)
G3 (Ki67 ≥ 20%)	2 (13.3)
Ki67 index at first diagnosis	
Ki67, mean ± SD (%)	7.7 ± 8.6
Ki67, median (%)	2.0
Ki67, complete distribution (%)	0,6, 1, 1, 1, 2, 2, 2, 2, 5, 5, 14, 15, 20, 20, 25
Primary tumor site, n (%)	
Ileum	12 (80.0)
Lung	2 (13.3)
Gastral	1 (6.7)

**Table 2 cancers-17-03907-t002:** Overview of systemic and local treatment regimens across therapy lines in patients with cardiac metastases from neuroendocrine neoplasms (n = 15). The table summarizes the sequential application of surgical approaches, somatostatin analogs (SSA), peptide receptor radionuclide therapy (PRRT), chemotherapy regimens, targeted therapies, radiotherapy, and interventional procedures (e.g., SIRT, TACE), over up to 12 treatment lines. Values are presented as absolute numbers with corresponding percentages. Abbreviations: CAPTEM = capecitabine/temozolomide, FOLFIRI = folinic acid, fluorouracil (5FU), and irinotecan, FOLFOX = folinic acid, fluorouracil (5FU), and oxaliplatin, n = number, PRRT = peptide receptor radionuclide therapy, SSA = somatostatin analogs, SIRT = selective internal radiation therapy, TACE = transarterial chemoembolization, TKI = tyrosine kinase inhibitor, % = percentage.

Treatment Schemes	Patients
1st line therapy regimes, n (%)	n = 15 (100)
Surgery	14 (93.3)
SSA	1 (6.7)
2nd line therapy regimes, n (%)	n = 15 (100)
Surgery	3 (20.0)
SSA	10 (66.7)
Platin/Etoposide	1 (6.7)
SIRT	1 (6.7)
3rd line therapy regimes, n (%)	n = 13 (100)
Surgery	1 (7.7)
SSA	3 (23.3)
CAPTEM	1 (7.7)
Platin/Etoposide	1 (7.7)
PRRT	3 (23.3)
Irradiation	2 (15.4)
Everolimus	1 (7.7)
TKI	1 (7.7)
4th line therapy regimes, n (%)	n = 8 (100)
Surgery	1 (12.5)
SSA	2 (25.0)
PRRT	4 (50.0)
Irradiation	1 (12.5)
Further therapy regimes (6th–12th)	
Surgery	
Irradiation	
SIRT	
SSA	
PRRT	
CAPTEM	
FOLFIRI	
FOLFOX	
Everolimus	
Platin/Etoposide	
Nivolumab/Ipilimumab	

## Data Availability

The raw data supporting the conclusions of this article will be made available by the authors, upon reasonable request.

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
