# Peer review of "Cardiac Metastases in Neuroendocrine Neoplasms: A Single-Center Experience of Clinical Characteristics and Outcomes"

_cancers, 2025, doi:10.3390/cancers17243907_

Round 1
Reviewer 1 Report
Comments and Suggestions for Authors
This study provides one of the few focused evaluations of cardiac metastases in NEN, leveraging a large single-center cohort where SSTR PET/CT is routinely incorporated. The work adds value by contextualizing CM within overall metastatic burden and exploring therapy adjustments, especially PRRT, after CM detection. Although retrospective and small, the dataset captures clinically silent cases that are typically missed, offering an important reference point for future prospective multicenter studies.
- The manuscript states that 1,201 NEN patients were screened, but the workflow for excluding patients who lacked adequate cardiac imaging is not clearly described. How many did or did not undergo SSTR PET/CT, cardiac MRI, or echocardiography? Without this information, the true prevalence estimate (1.25%) may be inaccurate or biased. Please detail inclusion/exclusion logic and the minimal imaging requirements for detecting CM.
- Although SSTR PET/CT detected most lesions, the manuscript does not quantify its incremental diagnostic value over CT/MRI. How many lesions were missed or discordant between modalities?
- Why was MTV included, and what clinical question does it answer? If MTV did not correlate with symptoms or survival, this should be explicitly discussed.
- The discussion mentions environmental and mechanical constraints of cardiac tissue, but the link between these biological mechanisms and the study’s observed findings is not fully articulated. How do the authors reconcile small CM size with absence of biomarker elevation or arrhythmias in most patients? This needs a stronger, mechanistically grounded explanation.
- The final statement suggests that CM is “an indicator of metastatic spread rather than a determinant of survival.” Given the small sample size and absence of controlled comparisons, this conclusion should be softened or framed as a hypothesis generating observation, not definitive evidence.
- Table 2 is oversized and difficult to navigate. Consider condensing or summarizing treatment lines >5 into grouped categories.
- The PET/CT, MRI, and echocardiography images lack scale bars, which are critical for contextualizing lesion size.
Author Response
Reviewer 1:
This study provides one of the few focused evaluations of cardiac metastases in NEN, leveraging a large single-center cohort where SSTR PET/CT is routinely incorporated. The work adds value by contextualizing CM within overall metastatic burden and exploring therapy adjustments, especially PRRT, after CM detection. Although retrospective and small, the dataset captures clinically silent cases that are typically missed, offering an important reference point for future prospective multicenter studies.
- The manuscript states that 1,201 NEN patients were screened, but the workflow for excluding patients who lacked adequate cardiac imaging is not clearly described. How many did or did not undergo SSTR PET/CT, cardiac MRI, or echocardiography? Without this information, the true prevalence estimate (1.25%) may be inaccurate or biased. Please detail inclusion/exclusion logic and the minimal imaging requirements for detecting CM.
We thank the reviewer for pointing us towards this important issue. Between January 2017 and May 2025, a total of 1,201 NEN patients were diagnosed and/or treated at ENETS Center of Excellence Tuebingen, Germany. All of these patients were screened for whether they had undergone SSTR PET/CT, cardiac MRI, and echocardiography. Most - but not all - patients received SSTR PET/CT. The reports of the SSTR PET/CT were screened using predefined keywords (“cardiac,” “ventricle,” “atrium,” and “septum”) to identify patients in whom cardiac metastases had been described. A substantially smaller number of patients had undergone cardiac MRI or echocardiography. These examinations were typically performed when carcinoid heart disease was suspected or present, or when another pre-existing cardiac pathology was known. The cardiac MRI and echocardiography reports were likewise screened for the evidence of cardiac metastases. In cases with ambiguous findings, image reinterpretation was performed in consultation with nuclear medicine, radiology, or cardiology. In total, 15 patients were identified with cardiac metastases. The remaining 1,186 patients were excluded from the study because cardiac metastases were not reported on SSTR PET/CT, cardiac MRI, or echocardiography, or because these examinations had not been performed. Accordingly, the resulting prevalence of cardiac metastases of 1.25% in this cohort must be interpreted with caution, as not all 1,201 patients underwent all three imaging modalities (SSTR PET/CT, cardiac MRI, and echocardiography).
To reflect this limitation, the manuscript was amended accordingly:
Methods (page 3, lines 117 ff.):
Therefore, multimodal imaging results such as echocardiography, CT, cardiac MRI and SSTR PET/CT were screened using predefined keywords (“cardiac”, “ventricle”, “atrium” and “septum”). In cases with ambiguous findings, image reinterpretation was performed in consultation with nuclear medicine, radiology, or cardiology.
Discussion (page 14, lines 449 ff.):
Notably, the observed 1.25% prevalence of CM in this cohort should be interpreted cautiously, since not all 1,201 patients received all three imaging modalities (SSTR PET/CT, cardiac MRI, and echocardiography).
- Although SSTR PET/CT detected most lesions, the manuscript does not quantify its incremental diagnostic value over CT/MRI. How many lesions were missed or discordant between modalities?
We thank the reviewer for raising this important issue. We agree that quantifying the incremental diagnostic value of SSTR PET/CT is essential for interpreting the imaging findings. Only one patient had CM initially suspected on CT, which then prompted a confirmatory cardiac MRI that led to the diagnosis. Another symptomatic patient received the initial CM diagnosis via cMRI after echocardiography. All remaining thirteen patients were diagnosed de novo by SSTR PET/CT. Across all patients with CM, four individuals underwent cMRI. In three of these cases, the number and location of lesions detected were identical between cMRI and SSTR PET/CT; in two patients, cMRI was performed first and subsequently confirmed by SSTR PET/CT, whereas in one patient SSTR PET/CT was obtained first and cMRI corroborated the findings. In the one patient that received initially transthoracic echocardiography, CM diagnosis was made externally on the basis of a single metastasis visualized on cMRI. However, SSTR PET/CT was performed after two cycles of PRRT in 2024, and from that time onward no cardiac lesions were detectable on SSTR PET/CT. No further cMRI was carried out to determine whether the initially observed cardiac lesion had also resolved on cMRI. Regarding CT, eight patients underwent either thoracic CT or whole-body contrast-enhanced CT during the diagnostic pathway; among these, only three showed a visible cardiac lesion on CT. SSTR PET/CT identified cardiac metastases in 14 of 15 patients; the single undetected case corresponds to the patient described above.
To reflect this issue, the manuscript was amended accordingly:
Methods (page 5, lines 143 ff.):
One patient had CM first suspected on CT and confirmed by cMRI; another symptomatic patient was diagnosed by cMRI after echocardiography. The remaining 13 patients were diagnosed de novo by SSTR PET/CT. Four patients underwent cMRI, and in three of them lesion number and location matched SSTR PET/CT findings. In two cases cMRI preceded PET/CT, and in one PET/CT preceded cMRI. The single patient initially evaluated with echocardiography received an external cMRI-based diagnosis of a solitary cardiac lesion, which was no longer detectable on SSTR PET/CT after two PRRT cycles; no follow-up cMRI was performed. Eight patients had thoracic or whole-body contrast CT, but only three showed cardiac lesions. Overall, SSTR PET/CT detected CM in 14 of 15 patients, with the only missed case being the patient described above.
- Why was MTV included, and what clinical question does it answer? If MTV did not correlate with symptoms or survival, this should be explicitly discussed.
We thank the reviewer for this rightful and important comment. As described by Trautwein et al. 2023 MTV in combination with chromogranin A can represent a significant prognostic factor for therapeutic outcome and long-term OS in NEN patients receiving PRRT (1). In the present study on cardiac metastases in patients with NEN, a central question asked why cardiac metastases are symptomatic in some cases and asymptomatic in others. Furthermore, the prognostic impact of cardiac metastasis on overall survival was of major interest. To investigate this, symptomatic and asymptomatic patients were evaluated with respect to the localization and number of cardiac metastases. Another question was whether total tumor volume or cardiac tumor volume contributes to the presence of symptoms in patients with CM. Because molecular tumor volume provides more information than macroscopic tumor volume - particularly regarding somatostatin receptor expression - both total and cardiac MTV were assessed in our study. Given that only four of the 15 patients exhibited symptoms related to their cardiac metastases, no association could be established between symptomatology and the localization, number, or MTV of CM. Nevertheless, we consider it plausible that in a larger patient cohort, a relationship between symptomatology and the localization, number, and MTV of CM could emerge, potentially leading to differences in OS. A multicenter investigation including a larger cohort of NEN patients with cardiac metastases is planned, in which the associations between localization, number, and MTV of CM, symptomatology, and OS will be re-evaluated.
References:
- Long-term prognostic factors for PRRT in neuroendocrine tumors. Trautwein NF, Schwenck J, Jacoby J, Reischl G, Fiz F, Zender L, Dittmann H, Hinterleitner M, la Fougère C. Front Med (Lausanne). 2023 Jun 9;10:1169970. doi: 10.3389/fmed.2023.1169970. eCollection 2023. PMID: 37359009
To reflect this point, the manuscript was amended accordingly:
Discussion (page12-13, lines 352 ff.):
Although MTV combined with chromogranin has been established as prognostic marker in NENs (PMID: 37359009), with only four symptomatic patients, no associations could be identified between symptoms and CM localization, number, or MTV; though such relationships, and corresponding OS differences, may emerge in a larger cohort.
- The discussion mentions environmental and mechanical constraints of cardiac tissue, but the link between these biological mechanisms and the study’s observed findings is not fully articulated. How do the authors reconcile small CM size with absence of biomarker elevation or arrhythmias in most patients? This needs a stronger, mechanistically grounded explanation.
We thank the reviewer for this insightful comment and fully agree with this statement. Although the observation that CM frequently remain clinically silent is repeatedly reported in the literature (1–4), the underlying mechanisms are not addressed in detail. To our knowledge, no study systematically evaluated the question why CM are predominantly indolent. We therefore propose several plausible mechanistic considerations that may contribute to this phenomenon.
First, and in line with the discussion presented earlier in the manuscript, the unique myocardial microenvironment – characterized by its distinct mechanical, metabolic, and immunological properties – appears to limit the growth pattern and clinical expression of CM, and hemodynamically relevant presentations are uncommon.
Second, we hypothesize additional mechanisms that may account for the frequently non-arrhythmogenic behavior of CM. We hypothesize that CM in NEN typically grow slowly and infiltratively, replacing myocardial tissue in a diffuse and non-compressive pattern without necessarily inducing ischemia. Also, the tumor cells themselves are electrically inert, as they lack the excitable ion-channel repertoire of cardiomyocytes (e.g., fast voltage-gated sodium channels and inward-rectifier potassium channels) (5). In addition, the myocardium exhibits substantial conduction redundancy through a dense and highly organized gap-junction network, which allows electrical activation to bypass non-conductive tumor-infiltrated regions as long as transition zones remain smooth and continuous (6,7).
These features stand in clear contrast to the well-characterized arrhythmogenic substrate of myocardial fibrosis. Fibrosis – especially when caused by infarction – creates abrupt conduction barriers that are highly arrhythmogenic even when spatially limited (6,7). Post-infarction border zones contain heterogeneous mixtures of surviving myocytes interspersed with fibrosis, creating intermediate electrophysiological properties with slow-conducting channels and regions of unidirectional block that facilitate re-entrant arrhythmias (6–8). Fibrotic remodeling also increases myocardial stiffness, and impairs diastolic relaxation, whereas it can be hypothesized that infiltrative CM generally do not compromise coronary perfusion or ventricular compliance as much until very advanced disease stages. This might provide plausible explanations for the predominantly silent clinical behavior of CM in NEN.
References:
- Bussani, R.; De‐Giorgio, F.; Abbate, A.; Silvestri, F. Cardiac Metastases. J. Clin. Pathol. 2007, 60, 27–34, doi:10.1136/jcp.2005.035105.
- Hamza, M.; Manasrah, N.; Patel, N.N.; Sattar, Y.; Patel, B. A Systematic Review and Meta-Analysis of Prevalence and Outcomes of Cardiac Metastasis of Neuroendocrine Malignancies. Am. J. Cardiol. 2023, 194, 86–92, doi:10.1016/j.amjcard.2023.02.011.
- Tyebally, S.; Chen, D.; Bhattacharyya, S.; Mughrabi, A.; Hussain, Z.; Manisty, C.; Westwood, M.; Ghosh, A.K.; Guha, A. Cardiac Tumors. JACC CardioOncology 2020, 2, 293–311, doi:10.1016/j.jaccao.2020.05.009.
- Liu, M.; Armeni, E.; Navalkissoor, S.; Davar, J.; Sullivan, L.; Leigh, C.; O’Mahony, L.F.; Hayes, A.; Mandair, D.; Chen, J.; et al. Cardiac Metastases in Patients with Neuroendocrine Tumours: Clinical Features, Therapy Outcomes, and Prognostic Implications. Neuroendocrinology 2020, 111, 907–924, doi:10.1159/000510444.
- Anderson, K.J.; Cormier, R.T.; Scott, P.M. Role of Ion Channels in Gastrointestinal Cancer. World J. Gastroenterol. 2019, 25, 5732–5772, doi:10.3748/wjg.v25.i38.5732.
- Kléber, A.G.; Rudy, Y. Basic Mechanisms of Cardiac Impulse Propagation and Associated Arrhythmias. Physiol. Rev. 2004, 84, 431–488, doi:10.1152/physrev.00025.2003.
- De Bakker, J.M.; Van Capelle, F.J.; Janse, M.J.; Tasseron, S.; Vermeulen, J.T.; De Jonge, N.; Lahpor, J.R. Slow Conduction in the Infarcted Human Heart. “Zigzag” Course of Activation. Circulation 1993, 88, 915–926, doi:10.1161/01.CIR.88.3.915.
- Ursell, P.C.; Gardner, P.I.; Albala, A.; Fenoglio, J.J.; Wit, A.L. Structural and Electrophysiological Changes in the Epicardial Border Zone of Canine Myocardial Infarcts during Infarct Healing. Circ. Res. 1985, 56, 436–451, doi:10.1161/01.RES.56.3.436.
To reflect this issue, the manuscript was amended accordingly:
Discussion (page 14, lines 420 ff.):
Beyond these microenvironmental influences, we further hypothesize that CM in NEN typically grow slowly and infiltratively, replacing myocardial tissue in a diffuse and non-compressive pattern without necessarily inducing ischemia. Also, the tumor cells themselves are electrically inert, as they lack the excitable ion-channel repertoire of cardiomyocytes (e.g., fast voltage-gated sodium channels and inward-rectifier potassium channels) (PMID: 31636470). In addition, the myocardium exhibits substantial conduction redundancy through a dense and highly organized gap-junction network, which allows electrical activation to bypass non-conductive tumor-infiltrated regions as long as transition zones remain smooth and continuous (PMID: 15044680, PMID: 8353918).These features stand in clear contrast to the well-characterized arrhythmogenic substrate of myocardial fibrosis. Fibrosis, especially when caused by infarction, creates abrupt conduction barriers that are highly arrhythmogenic even when spatially limited(PMID: 15044680, PMID: 8353918). Post-infarction border zones contain heterogeneous mixtures of surviving myocytes interspersed with fibrosis, creating intermediate electrophysiological properties with slow-conducting channels and regions of unidirectional block that facilitate re-entrant arrhythmias (PMID: 15044680, PMID: 8353918, PMID: 3971515). Fibrotic remodeling also increases myocardial stiffness, and impairs diastolic relaxation, whereas it can be hypothesized that infiltrative CM generally do not compromise coronary perfusion or ventricular compliance as much until very advanced disease stages. This might provide plausible explanations for the predominantly silent clinical behavior of CM in NEN.
- The final statement suggests that CM is “an indicator of metastatic spread rather than a determinant of survival.” Given the small sample size and absence of controlled comparisons, this conclusion should be softened or framed as a hypothesis generating observation, not definitive evidence.
We thank the reviewer for pointing us towards this important issue. Therefore, the final statement was revised, respectively.
The manuscript was amended accordingly:
Conclusion (page 15, lines 460 ff.):
Despite the presence of cardiac metastases OS remained favorable in this small cohort of NEN patients. We therefore hypothesize that cardiac metastases may rather be an indicator of metastatic spread than a standalone diagnostic determinant of survival.
- Table 2 is oversized and difficult to navigate. Consider condensing or summarizing treatment lines >5 into grouped categories.
We thank the reviewer for this helpful comment. Table 2 was amended accordingly:
Table 2
|
TREATMENT SCHEMES |
PATIENTS |
|
|
|
|
1st line therapy regimes, n (%) |
n = 15 (100) |
|
Surgery |
14 (93.3) |
|
SSA |
1 (6.7) |
|
|
|
|
2nd line therapy regimes, n (%) |
n = 15 (100) |
|
Surgery |
3 (20.0) |
|
SSA |
10 (66.7) |
|
Platin/Etoposide |
1 (6.7) |
|
SIRT |
1 (6.7) |
|
|
|
|
3rd line therapy regimes, n (%) |
n = 13 (100) |
|
Surgery |
1 (7.7) |
|
SSA |
3 (23.3) |
|
CAPTEM |
1 (7.7) |
|
Platin/Etoposide |
1 (7.7) |
|
PRRT |
3 (23.3) |
|
Irradiation |
2 (15.4) |
|
Everolimus |
1 (7.7) |
|
TKI |
1 (7.7) |
|
|
|
|
4th line therapy regimes, n (%) |
n = 8 (100) |
|
Surgery |
1 (12.5) |
|
SSA |
2 (25.0) |
|
PRRT |
4 (50.0) |
|
Irradiation |
1 (12.5) |
|
|
|
|
5th line therapy regimes, n (%) |
n = 6 (100) |
|
SSA |
3 (50.0) |
|
TACE |
1 (16.7) |
|
Irradiation |
2 (33.3) |
|
|
|
|
Further therapy regimes (6th-12th) |
|
|
Surgery |
|
|
Irradiation |
|
|
SIRT |
|
|
SSA |
|
|
PRRT |
|
|
CAPTEM |
|
|
FOLFIRI |
|
|
FOLFOX |
|
|
Everolimus |
|
|
Platin/Etoposide |
|
|
Nivolumab/Ipilimumab |
|
- The PET/CT, MRI, and echocardiography images lack scale bars, which are critical for contextualizing lesion size.
We thank the reviewer for raising this point. A scale bar for PET/CT images was added to Figure 3A. In consultation with the radiology colleagues, it is uncommon to include scale bars in cMRI images; therefore, they were omitted in this case. The scales shown for transthoracic and transesophageal echocardiography in Figure 3C indicate increments of 1 cm per marker.
Figure 3 was amended accordingly.
Reviewer 2 Report
Comments and Suggestions for Authors
The manuscript addresses an important and not completely erexplored topic. The study is well-structured, includes clear objectives, and provides valuable clinical insights. However, few issues need attention before publication.
- The conclusion that cardiac metastases do not significantly impact survival may be misleading given to the small cohort. Maybe Authors could rephrase to emphasize that findings are hypothesis-generating rather than definitive.
- Figures are referenced but not fully explained in the text (e.g., Figure 4 panels E and F lack interpretation). Authors can add explanatory captions and integrate figure interpretation into the results section.
- Define all abbreviations at first use in the abstract (e.g., PRRT, SSTR).
- Several sentences are lengthy and repetitive. Minor grammatical corrections and abbreviation consistency are needed.
Several sentences are lengthy and repetitive. Minor grammatical corrections and abbreviation consistency are needed.
Author Response
Reviewer 2:
The manuscript addresses an important and not completely erexplored topic. The study is well-structured, includes clear objectives, and provides valuable clinical insights. However, few issues need attention before publication.
- The conclusion that cardiac metastases do not significantly impact survival may be misleading given to the small cohort. Maybe Authors could rephrase to emphasize that findings are hypothesis-generating rather than definitive.
We thank the reviewer for pointing us towards this important issue. The final statement was revised, respectively.
The manuscript was amended accordingly:
Conclusion (page 15, lines 460 ff.):
Despite the presence of cardiac metastases OS remained favorable in this small cohort of NEN patients. We therefore hypothesize that cardiac metastases may rather be an in-dicator of metastatic spread than a standalone diagnostic determinant of survival.
- Figures are referenced but not fully explained in the text (e.g., Figure 4 panels E and F lack interpretation). Authors can add explanatory captions and integrate figure interpretation into the results section.
We thank the reviewer for this insightful advice. We explained Figure 4 panels E and F in the text of the manuscript.
The manuscript was amended accordingly:
Results (page 9, lines 289 ff.):
Compared to patients without symptoms of CM, symptomatic patients showed comparable total and CM specific MTV volumes (Figure 4E, F).
- Define all abbreviations at first use in the abstract (e.g., PRRT, SSTR).
We thank the reviewer for raising this point. All abbreviations are now defined at first use in the abstract.
The manuscript was amended accordingly.
- Several sentences are lengthy and repetitive. Minor grammatical corrections and abbreviation consistency are needed.
We thank the reviewer for this rightful and important comment. The manuscript was checked for grammatical correctness and abbreviation consistency.
Reviewer 3 Report
Comments and Suggestions for Authors
Comment
This manuscript by Lewetag focused on a clinically relevant topic: cardiac metastases (CM) in neuroendocrine neoplasms (NEN). The authors present a retrospective single-center cohort of 15 CM cases out of 1,201 NEN patients, offering imaging characteristics, clinical presentation, treatment decisions, and survival outcomes. The study is clearly written, includes multimodal imaging examples, and provides useful descriptive data. The topic is relevant to clinicians managing NEN, especially with increasing use of somatostatin receptor PET/CT. I would suggest the publication of the manuscript in cancers after addressing the following comments:
- Does the retrospective study in this field be reported for the first time? If no, I suggest highlighting what new information in this dataset, beyond confirming previously reported prevalence and imaging patterns, or emphasizing unique strengths in this study compared to others.
- The manuscript relies on SSTR PET/CT detection, please provide details in specify PET scanner model, acquisition protocol, reconstruction algorithm in Materials and Methods
- Please provide figures in a high-resolution version (e.g., Figure 2-5).
- Some journal names in References part are given in abbreviated form while most of those are given in full. Please check the typos to make sure every paper you cited is given in the same form.

Author Response
Reviewer 3:
This manuscript by Lewetag focused on a clinically relevant topic: cardiac metastases (CM) in neuroendocrine neoplasms (NEN). The authors present a retrospective single-center cohort of 15 CM cases out of 1,201 NEN patients, offering imaging characteristics, clinical presentation, treatment decisions, and survival outcomes. The study is clearly written, includes multimodal imaging examples, and provides useful descriptive data. The topic is relevant to clinicians managing NEN, especially with increasing use of somatostatin receptor PET/CT. I would suggest the publication of the manuscript in cancers after addressing the following comments:
- Does the retrospective study in this field be reported for the first time? If no, I suggest highlighting what new information in this dataset, beyond confirming previously reported prevalence and imaging patterns, or emphasizing unique strengths in this study compared to others.
We thank the reviewer for pointing us towards this important issue. Our study is not the first to address CM in NEN; e.g. a notable earlier contribution was made by Liu et al. in 2020 (1). However, by modelling CM as a time-dependent covariate, our analysis provides a more refined evaluation of their potential impact on therapeutic decision-making. In addition, we systematically assess arrhythmia burden in relation to CM number, showing that arrhythmias correlate poorly with CM count itself. By examining a cohort with consistent SSTR PET/CT imaging, our study also offers a more accurate estimate of the true incidence of CM in the contemporary imaging era, thereby addressing the possible under-recognition in earlier cohorts with heterogeneous imaging modalities.
References:
- Liu, M.; Armeni, E.; Navalkissoor, S.; Davar, J.; Sullivan, L.; Leigh, C.; O’Mahony, L.F.; Hayes, A.; Mandair, D.; Chen, J.; et al. Cardiac Metastases in Patients with Neuroendocrine Tumours: Clinical Features, Therapy Outcomes, and Prognostic Implications.
To reflect this issue, the manuscript was amended accordingly:
Significance (page 1, lines 30 ff.):
By modeling cardiac metastases as a time-dependent covariate, our analysis offers a nuanced assessment of their influence on therapeutic decision-making. We demonstrate that arrhythmia burden does not correlate with the number of cardiac lesions, indicating that lesion count alone is an insufficient marker of electrophysiologic risk. Moreover, the use of consistent SSTR PET/CT imaging across the cohort enables a accurate diagnosis of cardiac metastases, helping to address the under-recognition reported in earlier studies that relied on heterogeneous imaging modalities.
- The manuscript relies on SSTR PET/CT detection, please provide details in specify PET scanner model, acquisition protocol, reconstruction algorithm in Materials and Methods.
We thank the reviewer for raising this issue. Two different tracers were used for SSTR PET imaging: [18F]SiTATE and [68Ga]HA-DOTATATE, as described previously [1-3]. Scans with [68Ga]Ga-HA-DOTATATE were performed on a short axial field of view scanner (Biograph mCT; Siemens Healthineers; acquisition continuous bed motion of 0.7 mm/s) 20 minutes after i.v. injection of 3 MBq/kg BW of the tracer. Scans with [18F]SiTATE were conducted either on a short axial field of view scanner (Biograph mCT; Siemens Healthineers; acquisition continuous bed motion of 0.7 mm/s) or on a long axial field of view scanner (Siemens Biograph Vision Quadra; Siemens Healthineers; acquisition time of 5 min per bed position) 90 minutes after i.v. injection of 2-3 MBq/kg BW of the tracer.
For the Siemens Biograph Vision Quadra PET reconstruction was performed according to the standard clinical reconstruction protocol, with an Ordinary-Poisson Ordered-Subsets Expectation-Maximization algorithm (OP-OSEM 4 iterations, 5 subsets; Gaussian filter 4 mm), as described previously [4]. For the Biograph mCT data were corrected for attenuation as well as scatter and reconstructed iteratively with OSEM3D (2 iterations, 21 subsets; Gaussian filter, 2 mm).
The definition of pathologic SSTR expression was performed in joint consensus of board-certified nuclear medicine physician and a physician with several years of experience in hybrid imaging.
References:
- Trautwein, N.F., et al., Low-activity [(18)F]-somatostatin receptor (SSTR) imaging using [(18)F]SiTATE on a long axial field-of-view PET/CT scanner. EJNMMI Phys, 2025. 12(1): p. 13.
- Trautwein, N.F., et al., Histologic Ex Vivo Validation of the [(18)F]SITATE Somatostatin Receptor PET Tracer. J Nucl Med, 2025.
- Trautwein, N.F., et al., Radiosensitizing Favors Response to Peptide Receptor Radionuclide Therapy in Patients With Highly Proliferative Neuroendocrine Malignancies: Preliminary Evidence From a Clinical Pilot Study. Clin Nucl Med, 2024.
- Calderón, E., et al., One-day dual-tracer examination in neuroendocrine neoplasms: a real advantage of low activity LAFOV PET imaging. Eur J Nucl Med Mol Imaging., 2025.
To reflect this issue, the manuscript was amended accordingly:
Methods (page 6, lines 169 ff.):
PET acquisition and imaging analysis:
Two different tracers were used for SSTR-PET imaging [18F]SiTATE and [68Ga]HA-DOTATATE, as described previously (PMID: 39907960, PMID: 40404394, PMID: 38271237). Scans with [68Ga]Ga-HA-DOTATATE were performed on a short axial field of view scanner (Biograph mCT; Siemens Healthineers; acquisition continuous bed motion of 0.7 mm/s) 20 minutes after i.v. injection of 3 MBq/kg BW of the tracer.
Scans with [18F]SiTATE were conducted either on a short axial field of view scanner (Biograph mCT; Siemens Healthineers; acquisition continuous bed motion of 0.7 mm/s) or on a long axial field of view scanner (Siemens Biograph Vision Quadra; Siemens Healthineers; acquisition time of 5 min per bed position) 90 minutes after i.v. injection of 2-3 MBq/kg BW of the tracer.
For the Siemens Biograph Vision Quadra PET reconstruction was performed according to the standard clinical reconstruction protocol, with an Ordinary-Poisson Ordered-Subsets Expectation-Maximization algorithm (OP-OSEM 4 iterations, 5 subsets; Gaussian filter 4 mm), as described previously (PMID: 39883139).For the Biograph mCT data were corrected for attenuation as well as scatter and reconstructed iteratively with OSEM3D (2 iterations, 21 subsets; Gaussian filter, 2 mm).
The definition of pathologic SSTR expression was performed in joint consensus of board-certified nuclear medicine physician and a physician with several years of experience in hybrid imaging.
- Please provide figures in a high-resolution version (e.g., Figure 2-5).
We thank the reviewer for this rightful and important comment. All figures are now provided in a high-resolution version.
The Graphical Abstract, as well as Figures 1, 2, 3, 4, 5 and the Supplementary Figure 1-2 were amended accordingly.
- Some journal names in References part are given in abbreviated form while most of those are given in full. Please check the typos to make sure every paper you cited is given in the same form
We thank the reviewer for raising this issue. We checked the References part of the manuscript for typos and correct citation of each paper.
Reviewer 4 Report
Comments and Suggestions for Authors
This study focuses on the rare CM of NENs, a topic that holds clinical significance. The study design, which screened 15 CM patients from an initial single-center cohort of 1,201 cases, is targeted, and the results provide references for clinical diagnosis and treatment. However, the manuscript has deficiencies in several key sections and requires substantial supplementation and improvement before undergoing re-review.
- Please clarify the exclusion criteria, supplement the specific reasons and proportions of excluded cases in the initial cohort of 1,201 cases, and enhance the representativeness of the study cohort.
- Please supplement the technical parameters of somatostatin receptor positron emission tomography/computed tomography (SSTR PET/CT), and clarify whether the criteria for determining "pathological SSTR expression" have undergone standardized verification by the central laboratory.
- Please elaborate on the specific PRRT regimens, and specify the consistency of treatment among different patients as well as the rationale for individualized adjustments.
- Please supplement the monitoring indicators and results of treatment-related adverse reactions to verify the conclusion that "PRRT is well-tolerated".
- Please improve Table 1: supplement the complete age range (it is necessary to clarify whether 23–87 years represents the full range of the entire cohort) and the stratified distribution of Ki67 indices by tumor grade.
- Please supplement the statistical details of biomarker comparisons: include the p-values and confidence intervals for inter-group comparisons of CgA, NSE, and LDH between the time of CM diagnosis and the initial diagnosis of NENs, so as to clarify the statistical basis for the claim that "there is no significant increase".
- Please supplement the data on cardiac-specific biomarkers such as BNP and troponin in symptomatic and asymptomatic groups, and analyze their correlation with CM location and MTV.
- Please explain the potential mechanisms underlying the predilection of CM for the left ventricle, and link the imaging results of this study to known pathophysiological mechanisms.
9. Please explain the possible reasons for the "lack of correlation between symptoms and MTV", and conduct a stratified analysis based on the CM locations in the 4 symptomatic patients.
Author Response
Reviewer 4:
This study focuses on the rare CM of NENs, a topic that holds clinical significance. The study design, which screened 15 CM patients from an initial single-center cohort of 1,201 cases, is targeted, and the results provide references for clinical diagnosis and treatment. However, the manuscript has deficiencies in several key sections and requires substantial supplementation and improvement before undergoing re-review.
- Please clarify the exclusion criteria, supplement the specific reasons and proportions of excluded cases in the initial cohort of 1,201 cases, and enhance the representativeness of the study cohort.
We thank the reviewer for pointing us towards this important issue. Between January 2017 and May 2025, a total of 1,201 NEN patients were diagnosed and/or treated at ENETS Center of Excellence Tuebingen, Germany. All of these patients were screened for whether they had undergone SSTR PET/CT, cardiac MRI, and echocardiography. Most - but not all - patients received SSTR PET/CT. The reports of the SSTR PET/CT were screened using predefined keywords (“cardiac,” “ventricle,” “atrium,” and “septum”) to identify patients in whom cardiac metastases had been described. A substantially smaller number of patients had undergone cardiac MRI or echocardiography. These examinations were typically performed when carcinoid heart disease was suspected or present, or when another pre-existing cardiac pathology was known. The cardiac MRI and echocardiography reports were likewise screened for the evidence of cardiac metastases. In cases with ambiguous findings, image reinterpretation was performed in consultation with nuclear medicine, radiology, or cardiology. In total, 15 patients were identified as having cardiac metastases on imaging. The remaining 1,186 patients were excluded from the study because cardiac metastases were not reported on SSTR PET/CT, cardiac MRI, or echocardiography, or because these examinations had not been performed. Accordingly, the resulting prevalence of cardiac metastases of 1.25% in this cohort must be interpreted with caution, as not all 1,201 patients underwent all three imaging modalities (SSTR PET/CT, cardiac MRI, and echocardiography).
To reflect this issue, the manuscript was amended accordingly:
Methods (page 3, lines 117 ff.):
Therefore, multimodal imaging results such as echocardiography, CT, cardiac MRI and SSTR PET/CT were screened using predefined keywords (“cardiac”, “ventricle”, “atrium” and “septum”). In cases with ambiguous findings, image reinterpretation was performed in consultation with nuclear medicine, radiology, or cardiology.
Discussion (page 14, lines 449 ff.):
Notably, the observed 1.25% prevalence of CM in this cohort should be interpreted cautiously, since not all 1,201 patients received all three imaging modalities (SSTR PET/CT, cardiac MRI, and echocardiography).
- Please supplement the technical parameters of somatostatin receptor positron emission tomography/computed tomography (SSTR PET/CT), and clarify whether the criteria for determining "pathological SSTR expression" have undergone standardized verification by the central laboratory.
We thank the reviewer for raising this issue. Two different tracers were used for SSTR PET imaging: [18F]SiTATE and [68Ga]HA-DOTATATE, as described previously [1-3]. Scans with [68Ga]Ga-HA-DOTATATE were performed on a short axial field of view scanner (Biograph mCT; Siemens Healthineers; acquisition continuous bed motion of 0.7 mm/s) 20 minutes after i.v. injection of 3 MBq/kg BW of the tracer. Scans with [18F]SiTATE were conducted either on a short axial field of view scanner (Biograph mCT; Siemens Healthineers; acquisition continuous bed motion of 0.7 mm/s) or on a long axial field of view scanner (Siemens Biograph Vision Quadra; Siemens Healthineers; acquisition time of 5 min per bed position) 90 minutes after i.v. injection of 2-3 MBq/kg BW of the tracer.
For the Siemens Biograph Vision Quadra PET reconstruction was performed according to the standard clinical reconstruction protocol, with an Ordinary-Poisson Ordered-Subsets Expectation-Maximization algorithm (OP-OSEM 4 iterations, 5 subsets; Gaussian filter 4 mm), as described previously [4]. For the Biograph mCT data were corrected for attenuation as well as scatter and reconstructed iteratively with OSEM3D (2 iterations, 21 subsets; Gaussian filter, 2 mm).
The definition of pathologic SSTR expression was performed in joint consensus of board-certified nuclear medicine physician and a physician with several years of experience in hybrid imaging.
References:
- Trautwein, N.F., et al., Low-activity [(18)F]-somatostatin receptor (SSTR) imaging using [(18)F]SiTATE on a long axial field-of-view PET/CT scanner. EJNMMI Phys, 2025. 12(1): p. 13.
- Trautwein, N.F., et al., Histologic Ex Vivo Validation of the [(18)F]SITATE Somatostatin Receptor PET Tracer. J Nucl Med, 2025.
- Trautwein, N.F., et al., Radiosensitizing Favors Response to Peptide Receptor Radionuclide Therapy in Patients With Highly Proliferative Neuroendocrine Malignancies: Preliminary Evidence From a Clinical Pilot Study. Clin Nucl Med, 2024.
- Calderón, E., et al., One-day dual-tracer examination in neuroendocrine neoplasms: a real advantage of low activity LAFOV PET imaging. Eur J Nucl Med Mol Imaging., 2025.
To reflect this issue, the manuscript was amended accordingly:
Methods (page 6, lines 169 ff.):
PET acquisition and imaging analysis:
Two different tracers were used for SSTR-PET imaging [18F]SiTATE and [68Ga]HA-DOTATATE, as described previously (PMID: 39907960, PMID: 40404394, PMID: 38271237). Scans with [68Ga]Ga-HA-DOTATATE were performed on a short axial field of view scanner (Biograph mCT; Siemens Healthineers; acquisition continuous bed motion of 0.7 mm/s) 20 minutes after i.v. injection of 3 MBq/kg BW of the tracer.
Scans with [18F]SiTATE were conducted either on a short axial field of view scanner (Biograph mCT; Siemens Healthineers; acquisition continuous bed motion of 0.7 mm/s) or on a long axial field of view scanner (Siemens Biograph Vision Quadra; Siemens Healthineers; acquisition time of 5 min per bed position) 90 minutes after i.v. injection of 2-3 MBq/kg BW of the tracer.
For the Siemens Biograph Vision Quadra PET reconstruction was performed according to the standard clinical reconstruction protocol, with an Ordinary-Poisson Ordered-Subsets Expectation-Maximization algorithm (OP-OSEM 4 iterations, 5 subsets; Gaussian filter 4 mm), as described previously (PMID: 39883139).For the Biograph mCT data were corrected for attenuation as well as scatter and reconstructed iteratively with OSEM3D (2 iterations, 21 subsets; Gaussian filter, 2 mm).
The definition of pathologic SSTR expression was performed in joint consensus of board-certified nuclear medicine physician and a physician with several years of experience in hybrid imaging.
- Please elaborate on the specific PRRT regimens, and specify the consistency of treatment among different patients as well as the rationale for individualized adjustments.
We thank the reviewer for raising this important point. PRRT was administered in accordance with the practical guidelines jointly issued by the European Association of Nuclear Medicine, and the Society of Nuclear Medicine and Molecular Imaging, following the NETTER-1 protocol (1,2). Each treatment cycle consisted of an intravenous injection of 7400 MBq ± 10% [177Lu]Lu-HA-DOTATATE or [177Lu]Lu-DOTATATE, accompanied by an amino acid infusion for renal protection. One patient received two cycles of PRRT with 3000 ± 10% MBq [90Y]Y-DOTATOC, likewise accompanied by an amino acid infusion for renal protection. Overall, patients underwent between 2 and 8 PRRT cycles. [177Lu]Lu-HA-DOTATATE was prepared according to good manufacturing practice and the German Medicinal Products Act (AMG § 13 2b). [177Lu]Lu-DOTATATE was provided by Novartis. In 2 patients a radiosensitizer regimen was administered, as previously described (3).
References:
- Bodei, L., et al., The joint IAEA, EANM, and SNMMI practical guidance on peptide receptor radionuclide therapy (PRRNT) in neuroendocrine tumours. Eur J Nucl Med Mol Imaging, 2013. 40(5): p. 800-16.
- Strosberg, J., et al., Phase 3 Trial of (177)Lu-Dotatate for Midgut Neuroendocrine Tumors. N Engl J Med, 2017. 376(2): p. 125-135.
- Trautwein, N.F., et al., Radiosensitizing Favors Response to Peptide Receptor Radionuclide Therapy in Patients With Highly Proliferative Neuroendocrine Malignancies: Preliminary Evidence From a Clinical Pilot Study. Clin Nucl Med, 2024.
To reflect this issue, the manuscript was amended accordingly:
Methods (page 6, lines 188 ff.):
2.4 PRRT treatment modalities
PRRT was administered in accordance with the practical guidelines jointly issued by the European Association of Nuclear Medicine, and the Society of Nuclear Medicine and Molecular Imaging, following the NETTER-1 protocol (PMID: 23389427, PMID: 28076709). Each treatment cycle consisted of an intravenous injection of 7400 MBq ± 10% [177Lu]Lu-HA-DOTATATE or [177Lu]Lu-DOTATATE, accompanied by an amino acid infusion for renal protection. One patient received two cycles of PRRT with 3000 ± 10% MBq [90Y]Y-DOTATOC, likewise accompanied by an amino acid infusion for renal protection. Overall, patients underwent between 2 and 8 PRRT cycles. [177Lu]Lu-HA-DOTATATE was prepared according to good manufacturing practice and the German Medicinal Products Act (AMG § 13 2b). [177Lu]Lu-DOTATATE was provided by Novartis [7]. In 2 patients a radiosensitizer regimen was administered, as previously described (PMID: 38271237).
- Please supplement the monitoring indicators and results of treatment-related adverse reactions to verify the conclusion that "PRRT is well-tolerated".
We thank the reviewer for pointing us towards this important issue. To substantiate the statement that PRRT was well tolerated, we reviewed all available safety data, laboratory parameters, and follow-up documentation in patients who received PRRT after the diagnosis of CM. Among the 15 patients, eight underwent PRRT after CM detection. Two of these eight patients were treated at external institutions, therefore no toxicity data were available. In the remaining patients with complete safety data, treatment-related adverse events were generally mild. Reported hematologic toxicities comprised four cases of grade 1 anemia, two cases of grade 1 leukopenia, two cases of grade 1 neutropenia, one episode of grade 1 thrombocytopenia, and two episodes of grade 2 thrombocytopenia. Non-hematologic toxicity was limited to a single case of grade 1 creatinine elevation, without subsequent renal deterioration. Across all available structured reports, treating physicians in our institution consistently described PRRT as well tolerated, and no treatment discontinuations or hospitalizations attributable to toxicity were recorded in these five.
To reflect this issue, the manuscript was amended accordingly:
Methods (page 6, lines 201 ff.):
Across all available structured reports, PRRT was as well tolerated, and no treatment discontinuations or hospitalizations attributable to toxicity were recorded. Reported hematologic toxicities comprised four cases of grade 1 anemia, two cases of grade 1 leukopenia, two cases of grade 1 neutropenia, one episode of grade 1 thrombocytopenia, and two episodes of grade 2 thrombocytopenia. Non-hematologic toxicity was limited to a single case of grade 1 creatinine elevation, without subsequent renal deterioration.
- Please improve Table 1: supplement the complete age range (it is necessary to clarify whether 23–87 years represents the full range of the entire cohort) and the stratified distribution of Ki67 indices by tumor grade.
We thank the reviewer for this rightful and important comment.
Table 1 was amended accordingly.
- Please supplement the statistical details of biomarker comparisons: include the p-values and confidence intervals for inter-group comparisons of CgA, NSE, and LDH between the time of CM diagnosis and the initial diagnosis of NENs, so as to clarify the statistical basis for the claim that "there is no significant increase".
We thank the reviewer for pointing us towards this issue. Besides showing the p-values in the Suppl. Figure 1 we additionally added median and 95% CI for all respective groups (see Suppl. Figure 1). Moreover, this information was added to the manuscript.
Results (page 7, lines 238 ff.):
Compared to primary diagnosis, CgA level were slightly higher at timepoint of CM diagnosis (Median: 138 vs. 230 mg/L). However, due to high variability and small patient numbers, this trend was not statistically significant (p=0.2998). NSE (p=0.2693) as well as LDH (p=0.9337) levels showed no significant differences between primary diagnosis and diagnosis of CM.
- Please supplement the data on cardiac-specific biomarkers such as BNP and troponin in symptomatic and asymptomatic groups, and analyze their correlation with CM location and MTV.
We thank the reviewer for this important comment. We agree, that these comparisons provide further insight into the interplay of CM and cardiac function. We performed all requested analyses. However, due to the small patient size and the inconsistent measurement of, especially BNP and troponin, no significant trend was observed. To further investigate this, larger patient cohorts are needed.
We provide these data now in Suppl. Figure 2 and amended the manuscript as follows:
Results (page 7, lines 242 ff.):
To further assess the relationship between cardiac-specific biomarkers and CM, we examined BNP and troponin levels in both symptomatic and asymptomatic patients and analyzed their associations with CM localization and MTV values (Supplementary Fig. 2A–C). Although this analysis has the potential to provide additional insights into the interplay between CM and cardiac function, the limited availability of biomarker data and the small patient cohort unfortunately prevented more definitive conclusions.
- Please explain the potential mechanisms underlying the predilection of CM for the left ventricle, and link the imaging results of this study to known pathophysiological mechanisms.
We thank the reviewer for raising this point. We hypothesize that the predilection of CM for the left ventricle may reflect a predominantly hematogenous rather than lymphatic mechanism of spread. Hematogenous cardiac dissemination is documented in other tumor entities such as melanoma, lymphoma and sarcoma and associated with myocardial and endocardial metastases (1). In this context, the frequent infiltration of the left ventricle may be explained by its high myocardial perfusion, dense capillary network, and greater myocardial mass, which together may facilitate hematogenous microvascular tumor lodging and intramyocardial infiltration.
References:
- Burazor, I.; Aviel-Ronen, S.; Imazio, M.; Goitein, O.; Perelman, M.; Shelestovich, N.; Radovanovic, N.; Kanjuh, V.; Barshack, I.; Adler, Y. Metastatic Cardiac Tumors: From Clinical Presentation through Diagnosis to Treatment. BMC Cancer 2018, 18, 202, doi:10.1186/s12885-018-4070-x.
To reflect this issue, the manuscript was amended accordingly:
Discussion (page 14, lines 439 ff.):
The predilection of CM for the left ventricle may reflect a predominantly hematogenous rather than lymphatic mechanism of spread. Hematogenous cardiac dissemination is described in other tumor entities such as melanoma, lymphoma and sarcoma and associated with myocardial and endocardial metastases (PMID: 29463229). In this context, the frequent infiltration of the left ventricle may be explained by its high myocardial perfusion, dense capillary network, and greater myocardial mass, which together may facilitate hematogenous microvascular tumor lodging and intramyocardial infiltration.
- Please explain the possible reasons for the "lack of correlation between symptoms and MTV", and conduct a stratified analysis based on the CM locations in the 4 symptomatic patients.
We thank the reviewer for this comment. Even though our study provides interesting insights into CM in NEN, the patient size in these study remains small. As a result, powerful statistical analysis and definitive conclusions are not possible. Overall, the reviewer suggestions provide a well-thought and compelling framework for further multicenter analysis, investigating bigger patient cohorts.
Among the four symptomatic patients, two individuals presenting with atrial fibrillation and syncope exhibited CM located at the atrial septum and atrioventricular junction, potentially linking CM localization to disturbances in cardiac conduction. The remaining patients, who presented with atrial fibrillation and angina pectoris, displayed CM within the right and left ventricles. However, due to limited cohort size, definitive associations between specific cardiac symptoms and CM localization could not be established.
Round 2
Reviewer 1 Report
Comments and Suggestions for Authors
I find the current version technically sound, well-written, and suitable for publication. I therefore recommend the manuscript for acceptance in its present form.